# Some Factors Involved in the Success of Side Veneer Grafting of *Pinus engelmannii* Carr.

**Alberto Pérez-Luna [1], José Ángel Prieto-Ruíz [2], Javier López-Upton [3], Artemio Carrillo-Parra [4], Christian Wehenkel [4]** **, Jorge Armando Chávez-Simental [4] and José Ciro Hernández-Díaz [4,*]**

[1]  Programa Institucional de Doctorado en Ciencias Agropecuarias y Forestales, Universidad Juárez del Estado de Durango, Constitución 404 sur Zona Centro, Durango CP 34000, Mexico; aperez@ujed.mx
[2]  Facultad de Ciencias Forestales, Universidad Juárez del Estado de Durango, Río Papaloapan y Blvd. Durango S/N Col. Valle del Sur, Durango CP 34120, Mexico; jprieto@ujed.mx
[3]  Colegio de Postgraduados, Campus Montecillo, Carretera México-Texcoco Km 36.5, Montecillo CP 56230, Texcoco, Mexico; jlopezupton@gmail.com
[4]  Instituto de Silvicultura e Industria de la Madera, Universidad Juárez del Estado de Durango, Boulevard del Guadiana 501, Ciudad Universitaria, Torre de Investigación, Durango CP 34120, Mexico; acarrilloparra@ujed.mx (A.C.-P.); wehenkel@ujed.mx (C.W.); jorge.chavez@ujed.mx (J.A.C.-S.)
*   Correspondence: jciroh@ujed.mx; Tel.: +52-618-825-1886

**Abstract:** The establishment of clonal seed orchards is a viable option for the continuous production of improved seed of desired genotypes. Grafting is the main technique used to establish clonal seed orchards. The objective of this study was to examine how the geographic location and the age class of the donor trees of buds, the phenological status of the buds, and the anatomical characteristics of the scions and the rootstocks affect the survival and growth of *Pinus engelmannii* Carr. grafts. Scions were collected from two trees in each of three age classes (young, middle-aged, and old). Grafting was performed with buds in two physiological states (end of dormancy and beginning of sprouting). Cross-sections of the grafted organs were obtained for anatomical analysis. A Cox proportional hazards model was used to determine the effects of the variables that were considered. The age class of the scion donor trees, the total area of the cut surface of the scion, and the density of resin channels in the scions significantly affected ($p < 0.05$) survival of the grafts. By contrast, the physiological state of the buds and the other anatomical characteristics of the grafted organs did not significantly affect graft survival. In *P. engelmannii*, grafting was most effective when scions from middle-aged trees were used. Graft survival was enhanced by a small total area of the cut surface of the scion and low density of resin channels in the scions. The area of the cambium of the scions directly influenced growth of the grafts.

**Keywords:** donor tree; graft compatibility; resin channels; Cox model

## 1. Introduction

Grafting is a vegetative propagation technique that consists of joining parts of two different plants (a scion with a rootstock), which fuse together to form a callus, thus growing as a single plant [1–3]. Vegetative propagation by grafting allows the preserving of genetic resources of high commercial, ecological, and cultural value [2].

The provenance of the scion or barb has a greater effect on survival than the characteristics of the recipient rootstock when grafting *Pinus taeda* L. [4], and the quality of the rootstock that is used has significant effects on the survival and growth of the grafts [5]. Further, for the grafting of woody species, it is advisable to select scions and rootstocks of the highest possible quality [6]. Some authors

indicate that the best time to graft woody species is in the winter, i.e., between December and February in the northern hemisphere [7–9]. Nonetheless, it has also been noted in the literature that it is possible to graft conifers at any time of the year, if the temperature and humidity of the site where the grafts are maintained are controlled [10]. Several studies of conifers have demonstrated that grafting scions obtained from trees that are younger than 50 years of age increases the success of the grafting [1,7,11]. In addition, the geographical provenance of the scions and rootstocks may influence the success rate of the grafting [12].

The anatomical compatibility, between the rootstock and the scion carrying the buds, is a determining factor in the development of the graft [13,14]. Several authors have studied the process of fusion between the cambial zones of the scion and the rootstock used in grafts of *Eriobotrya japonica* (Thunb.) Lindl. [15], *Picea sitchensis* (Bong.) Carr. [16], and in the genus *Rhododendron* L. [17]. Three important stages of graft development are recognized in this process: callus formation, cambial differentiation, and cambial continuity (vascular tissue formation) [18].

In general, the compatibility of the rootstock and the scion is important in relation to the development of a graft. When the cambium areas of both organs are in contact (meristematic zone) and after callus formation, parenchyma cells are generated, which are necessary for photosynthesis, nutrient supply, and the protection of the xylem and the phloem [3]. Unfortunately, information regarding the effect of meristematic cell activity on grafts of *Pinus* species is scarce [19].

Some studies indicate that cell walls become diluted during callus formation, which enables the secretion of proteins and the formation of an agglomeration of meristematic cells, known as a "catalytic complex" [20,21]. The results of other studies indicate that it is impossible to prove the existence of this complex, given the difficulty in observing the changes that take place in the cell walls during callus formation [22,23]. It was concluded, in one study, that during the grafting of *Anacardium occidentale* L., the resin enables a temporary union of the grafts, and the number of resin channels may therefore have a positive effect, because the resin produced may act as an insulator against fungal attack and moisture loss [24]. However, several grafting studies that were carried out on *Anacardium occidentale* [24], *Trema orientalis* (L.) Bl., and *Julbernardia globiflora* Benth. [25], and on *Cucumis sativus* L. and *Cucurbita ficifolia* [26], undertook in-depth investigations of the processes of callus formation, and the results did not report any negative effect on callus formation due to the density of resin channels and resin production.

The grafting of conifers is useful for establishing asexual seed orchards (ASO), which are intended to be genetically improved germplasm reservoirs [27–29]. *Pinus engelmannii* Carr. is one of the most important coniferous species in northern Mexico, due to the quality and quantity of its wood and its dominant distribution in the forests of the Sierra Madre Occidental [30,31]. Notwithstanding, this species is widely used in reforestation programs and commercial forest plantations in northern Mexico [30]. Top cleft grafting and side veneer grafting are the grafting techniques that are most commonly used in conifer species [11,32]. The taxonomic affinity, genetic and anatomical compatibility, phenological status of the scion, vigor, health, age of the scion donor tree, and the characteristics of the rootstocks, as well as the temperature and humidity of the environment where the grafts are made and maintained, are reported as the most influential factors for grafting success [19,33–37]. However, specific information on these factors and their influence on graft survival in conifers is scarce [1,7,19,35,36].

Most of the commercial forest plantation programs in regions with climates ranging from semiarid with bimodal precipitation to temperate-subhumid, with most of the precipitation falling in the summer [38], have been made using *Pinus engelmannii*. Up until 2014, a total of 4600 ha had been planted in the state of Durango, Mexico. However, due to the lack of sources of seed with high genetic quality, the plants obtained and used in these plantations had genetic deficiencies which, in turn, had an impact on the low productivity of the planted material [39].

Therefore, the objective of the present study was to evaluate how the age class and the geographic location of the scion donor tree, as well as the anatomical characteristics of the scions and rootstocks and the phenological status of the buds, affect the survival and growth of grafts of *Pinus engelmannii*.

## 2. Materials and Methods

### 2.1. Scion Collection

The *Pinus engelmannii* scions were obtained from two seed production areas in the forests of the Sierra Madre Occidental in the state of Durango, Mexico. We selected two different provenances in this region, with the purpose of studying whether the results of grafting were similar or not. These provenances are separated by 28.8 km in longitude and 22.5 km in latitude. Provenance one (1) is located in the Otinapa region (24°03′16″ N and 105°00′39″ W) at an elevation of 2380 m, while provenance two (2) is located in the Ejido Llano Grande (23°51′56″ N and 105°16′12″ W) at an altitude of 2400 m. To test the age impact of the scion donor tree, we selected three ages in each studied location: juvenile (J) trees that were less than 20 years of age, middle-aged (M) trees having between 20 and 60 years of age, and old (V) trees with more than 60 years of age. The age of the trees was determined by counting the rings of each individual, from growth cores that were extracted with a Pressler drill. To obtain the scions, the trees were scaled by the researchers with the aid of spurs and a body harness. Twenty branchlets with scions were removed from the ends of branches that were located in the upper third of the crown of each tree. Finally, two collections were made in order to obtain scions with buds at two phenological stages, on 13–14 February 2017 (buds that were considered to be at the end of the dormant state, since the colder months are December and January) and on 20–21 March 2017 (buds that were visually at the beginning of sprouting).

The temperature conditions of the different provenances during the two months prior to each grafting season (phenological stages) are presented in Table 1. In the Otinapa region, an average precipitation of 11.5 and 20.0 mm occurred in the two months prior to the first and second scion collection, respectively, while in the Ejido Llano Grande region there was an average rainfall of 6.5 and 12.5 mm in the two months prior to each scion harvest.

**Table 1.** Temperature in the two scion collection sites of *Pinus engelmannii* Carr.

| Provenance | January–February (°C) | | | February–March (°C) | | |
|---|---|---|---|---|---|---|
| | Maximum | Minimum | Mean | Maximum | Minimum | Mean |
| Otinapa | 25.0 | −9.5 | 6.6 | 28.3 | −0.9 | 12.1 |
| Ejido Llano Grande | 22.5 | −10.5 | 6.1 | 24.4 | −0.7 | 11.0 |

### 2.2. Grafting

Grafting was performed in a greenhouse of dimensions $6 \times 8 \times 3$ m, covered with 720 micron caliber plastic, and with a 60% shading mesh placed in the upper part. The greenhouse is located at the Institute of Forestry and Wood Industry of the Juarez University of the State of Durango (ISIMA-UJED is the Spanish-language acronym). The grafting was carried out on the day after the scions of each provenance and phenological stage were collected. The treatments that were evaluated consisted of different combinations of provenance (2), age class of the donor tree (3) and phenology of the bud (2), resulting in 12 treatments replicated 10 times each (120 grafts in total) (Table 2). The statistical analysis of the results was based mainly on non-parametric techniques.

**Table 2.** Treatments evaluated for *Pinus engelmannii* Carr. grafting.

| Treatment | Provenance | Donor Tree | Phenological Stage of the Bud |
|---|---|---|---|
| P1-J- L<br>P1-M- L<br>P1-V- L | Provenance 1 | Young<br>Middle-aged<br>Old | End of dormancy |
| P1-J-B<br>P1-M- B<br>P1-V- B | | Young<br>Middle-aged<br>Old | Beginning of sprouting |
| P2-J- L<br>P2-M- L<br>P2-V- L | Provenance 2 | Young<br>Middle-aged<br>Old | End of dormancy |
| P2-J- B<br>P2-M- B<br>P2-V- B | | Young<br>Middle-aged<br>Old | Beginning of sprouting |

P1 = Provenance 1; P2 = Provenance 2; J = young tree; M = middle-aged tree; V = old tree; L = dormant; B = sprouting.

*Pinus engelmannii* plants of similar age (five to seven years), were obtained in July 2016 from the forest nurseries of the Municipal Government of Durango and the Valle del Guadiana Experimental Field of INIFAP, and they were used as rootstocks. These rootstock plants were moved from 3 L to 5 L bags in September 2016. The plants were irrigated every three days, and weeds were removed weekly. At the time of grafting, the average height of the rootstock plants was 80 cm, and the average diameter at the neck of the root was 3.5 cm. We used rootstocks of five to seven years, instead of using solely one age, because their variability in diameter allowed a better matching with the dimensions of the scions, which were also variable despite the fact that all of them had been taken from one-year branchlets. Notwithstanding, since the grafting was undertaken at the end of the winter season, and the age of the rootstock was not variable under evaluation, we considered the possible growth that the rootstock underwent between the two grafting dates to be negligible. Further, this latter variable was considered to be an independent variable with which to evaluate the success of grafting, because the phenological stage of the buds had obviously changed.

The 120 grafts were effected by using the side veneer technique, which consists of maintaining the complete rootstock (without removing the upper part). A longitudinal cut was made on the stem, and the surrounding bark was removed. The scion including the bud was inserted into the cut, with care being taken to not damage the cambium of the rootstock and, also, leaving a small slit in the bark at the lower end of the cut (Figure 1a). To obtain the scion, a cut of length 3 to 5 cm was made at the end that is opposite to the bud on each twig that was collected from the donor trees (Figure 1b). A wedge-shaped cut of approximately 0.5 cm was subsequently made in the lower part of the opposite side to the first cut of the scion (Figure 1c). The size and shape of the lateral cut that was made on the rootstock coincided with the size and shape of the long cut of the scion, in order to maximize the area of contact and the scarring of the graft. In addition, the diameters of the scion and the rootstock used were similar.

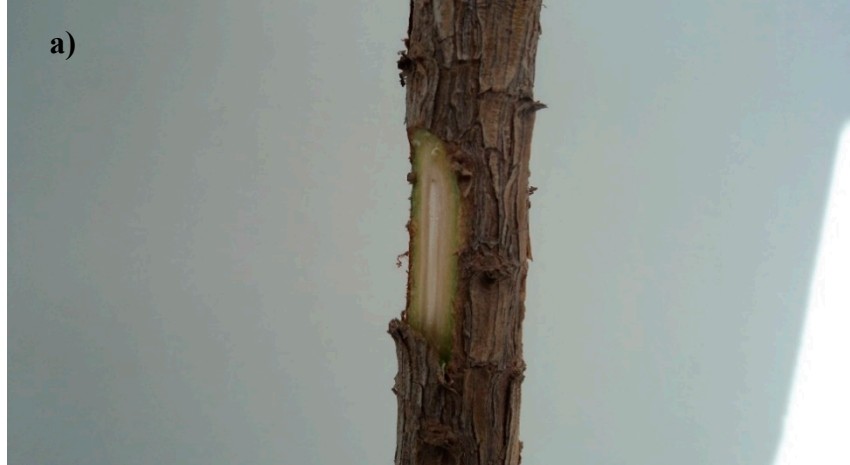

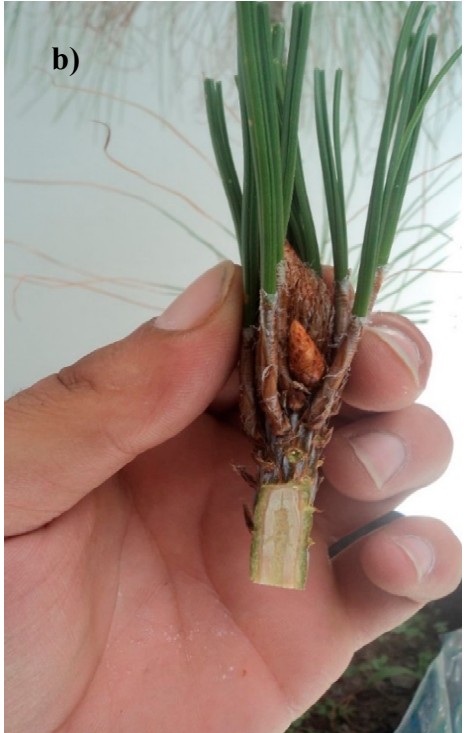
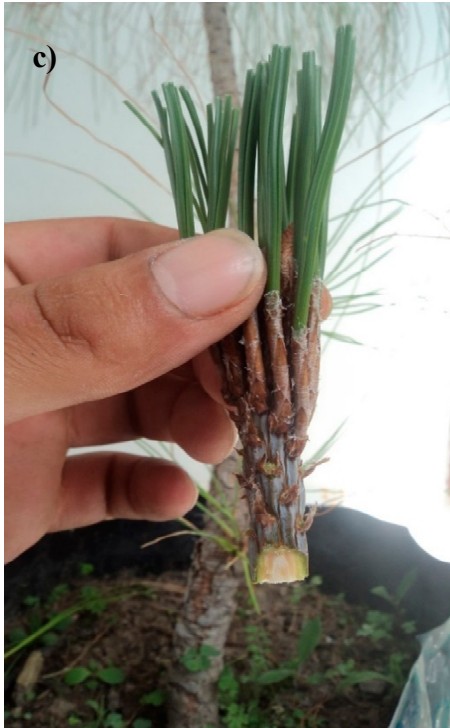

**Figure 1.** Cuts in rootstock and scion for grafting. (**a**) Longitudinal section in the rootstock with a slit at the lower end. (**b**) Long cut at the end of the scion that would be tied to the rootstock. (**c**) Small cut on the opposite side to the first cut, for insertion in the rootstock.

For grafting, the exposed parts were joined (Figure 2a). The area of contact between the scion and the rootstock was tied with rubber glass setting tape, to secure the joint. The grafted area was then sealed with water-soluble Captan® (Industrial Engineering, Mexico City, Mexico) fungicide (2 g $L^{-1}$) mixed with vinyl paint as adherent. A plastic bag containing water was placed around each graft to generate a wet microclimate (Figure 2b). Temperature and relative humidity (maximum, minimum and mean) were also recorded using a data logger (HOBO®, Onset Computer Corporation, Bourne, MA, USA) during the first six months after the grafting.

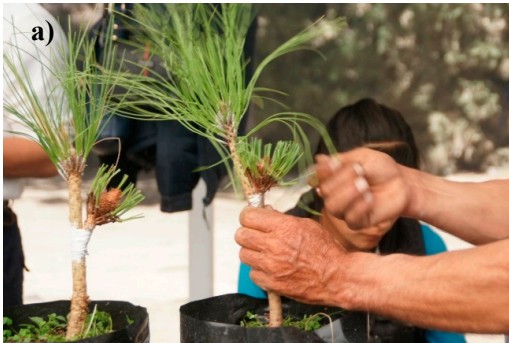 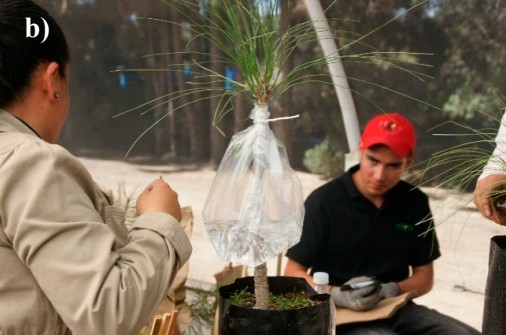

**Figure 2.** Final process of side veneer grafting. (**a**) Tying the graft. (**b**) Generating the microclimate.

### 2.3. Anatomical Analysis

An anatomical analysis of the grafted structures was undertaken. A transverse section of the cambium of each scion was obtained, and a cross-section of the respective grafted rootstock with each scion was also obtained for each treatment. One section sample was obtained for each scion and grafted rootstock. The section samples were polished using microfine sandpaper. These samples were first polished with sandpaper number 1000, by applying circular motions for five minutes until the cut was smoothed and reduced to 1.0 mm thickness. The samples were then polished with sandpaper numbers 1200, 1500, 2000, and 2500, for three minutes with each sandpaper, to obtain a flat and perfectly polished surface. The samples were examined using a Zeiss® (Carl Zeiss AG, Oberkochen, Germany) stereoscope and scanned using a Canon® (Canon Inc., Tokyo, Japan) EO5 RT6i camera with a resolution of 17.9 megapixels. Photographs were taken with a stereoscope approach of 0.8 to 1.0, according to the size of the sample, with a $4\times$ zoom lens fitted to the camera. The images were processed using Image-Pro Plus 4.5 software (Media Cybernetics, Inc., Rockville, MD, USA), working with a calibration of 0.00467 pixels/mm$^2$ for the scion samples and 0.00319 pixels/mm$^2$ for the rootstock samples. The following anatomical variables were quantified: the total area of the cut in the scion (TAS), total area of the rootstock (TAR), cambium area in the scion (CAS), cambium area in the rootstock (CAR), density of resin channels in the scion (DRCS), and density of resin channels in the rootstock (DRCR). The ratios between the anatomical variables in the scion and the rootstock were also analyzed. These ratios were determined by dividing the values of the anatomical variables of the scions by the values of the anatomical variables of the respective rootstocks (TAS/TAR, CAS/CAR, DRCS/DRCR) as a measure of the match between the areas of the organs. The surface area of the scions should be equal to, or slightly smaller than, that of the rootstock [1,32].

The maximum, minimum, and mean values of each analyzed anatomical variable of the 120 scions and 120 grafted rootstocks are presented in Table 3. The most variable parameter was the DRCR. All of the variables were normally distributed according to the Kolmogorov–Smirnov normality test.

**Table 3.** Results of the measurements of the anatomical variables of scions and rootstocks.

| Anatomical Variable | Maximum | Minimum | Mean | Coefficient of Variation |
|---|---|---|---|---|
| TAS (mm$^2$) | 170.9 | 31.2 | $87.1 \pm 26.0$ | 0.29 |
| TAR (mm$^2$) | 188.6 | 36.1 | $89.0 \pm 26.6$ | 0.29 |
| CAS (mm$^2$) | 91.9 | 17.9 | $52.2 \pm 14.2$ | 0.27 |
| CAR (mm$^2$) | 112.3 | 15.4 | $41.9 \pm 16.9$ | 0.40 |
| DRCS (Channels/mm$^2$) | 1.03 | 0.22 | $0.47 \pm 0.15$ | 0.31 |
| DRCR (Channels/mm$^2$) | 0.96 | 0.10 | $0.35 \pm 0.15$ | 0.42 |
| TAS/TAR | 1.16 | 0.85 | $0.97 \pm 0.08$ | 0.08 |
| CAS/CAR | 2.45 | 0.69 | $1.35 \pm 0.40$ | 0.29 |
| DRCS/DRCR | 3.10 | 0.63 | $1.52 \pm 0.56$ | 0.36 |

TAS = total area of the scion; TAR = total area of the rootstock; CAS = area of the cambium in the scion; CAR = area of the cambium in the rootstock; DRCS = density of resin channels in the scion, DRCR = density of resiniferous channels in the rootstock.

*2.4. Evaluated Variables and Statistical Analysis*

The variables evaluated were survival and total growth of the grafts, six months after grafting.

2.4.1. Graft Survival

Evaluations were carried out monthly for six months. Live grafts were coded as one (1) and dead grafts as zero (0). Analysis of variance (ANOVA) was used to evaluate the effect of the treatments on graft survival. The Kolmogorov–Smirnov normality test was applied to the anatomical variables that were evaluated. The relationships with pairs of variables were also compared by using the Student's *t*-test. The survival function of the Kaplan–Meier model, which determines the probability or risk of the occurrence of mortality in an individual at any given moment, was used to determine the dynamics of survival over the evaluation time (*t*) [40]. The survival function is defined as follows:

$$S(t) = p(T > t),\tag{1}$$

where *S(t)* is the probability of survival in a given time (*t*), and *p* is the probability of survival in the different time intervals during the evaluation; *T* is the total time of survival until the end of the evaluation, which must be greater than *t*, which is the time at any moment from the start of the evaluation.

The variables evaluated were the age class of the donor tree (DTA), provenance of the scion (SP), phenological status of the buds (PSB), total area of the cut in the scion (TAS), total area of the rootstock (TAR), area of the cambium in the scion (CAS), area of the cambium in the rootstock (CAR), density of resin channels in the scion (DRCS), and density of resin channels in the rootstock (DRCR). The respective effects were estimated by applying an analysis of a Cox proportional hazards model to the data [41,42], defined as follows:

$$h_i(t) = h_0(t)e^{(\beta_i x_{i1} + \cdots + \beta_k x_{ik})},\tag{2}$$

where $h_i$ is the risk of death of an individual in function of a given time (*t*), which is proportional to the product of the risk ratio at the beginning of the evaluation period or base risk, which depends on time ($h_0$), multiplied by an exponential function of *k* time-independent variables. However, the Cox proportional hazards model assumes that $h_0$ is an unknown or intangible value, which is why only the risk ratio (*HR*) was calculated on the basis of the known data of the variables and, for this reason, $h_0$ does not need to be known [41]. Therefore, the reduced model is as follows:

$$HR = \frac{h_i(t)}{h_0(t)} = e^{(\beta_1 x_{i1} + \cdots + \beta_k x_{ik})}.\tag{3}$$

The Cox reduced proportional hazards model considers the effect of the variables for defining estimators (β coefficients) which, together, represent the probability during the evaluation period that a given individual will die. If the estimator of some variable is negative, the probability of death decreases (survival increases) when the value of the independent variable increases and, if the estimator is positive, then the probability of death increases (survival decreases) when the independent variable increases.

In addition to the individual evaluation of each variable, the influence of the possible ratios between pairs of anatomical variables (TAS/TAR, CAS/CAR, DRCS/DRCR) and graft survival was evaluated.

Pearson correlation coefficients were also calculated in order to examine the relationship between the anatomical variables and the possible occurrence of multicollinearity, and to select explanatory variables for inclusion in the model. Stepwise regression was then used to enable the selection of the variables that contribute most to explaining the Cox proportional hazards model. The Kolmogorov–Smirnov normality test, the means comparison test, the Kaplan–Meier model,

the Cox proportional hazards model, and the stepwise regression were performed in the R software environment [43].

### 2.4.2. Model-Based Prediction of Graft Growth

Growth of the scions of the surviving grafts was measured six months after the grafts were established. Pearson correlation coefficients for growth of the grafts and the anatomical variables in the scions and the rootstocks were calculated. The data were fitted to four different models for predicting the growth of the grafts (Table 4). The Akaike criterion test (AIC) was used to identify the best-fit model; this was performed by using R software (R Core Team, Vienna, Austria) [43].

**Table 4.** Models used to predict the graft growth according to the anatomical characteristics of the scions.

| Model | Expression |
|-------|------------|
| Quadratic model | $y_i = \beta_0 + \beta_1 x_i + \beta_2 x_i^2$ |
| Logarithmic model | $y_i = \beta_0 + \beta_1 ln x_i$ |
| Linear model | $y_i = \beta_0 + \beta_1 x_i$ |
| Exponential model | $y_i = \beta_0 e^{\beta_1 x_i}$ |

## 3. Results

### 3.1. Wood Sections for Anatomical Analysis

The paired means test identified significant differences between the TAS vs. TAR variables. The TAS was generally lower than the TAR. Similarly, highly significant differences between the CAS vs. CAR and DRCS vs. DRCR variables were observed. In both cases, the variables corresponding to the scion were generally higher (Table 5).

**Table 5.** Student's *t*-test of paired anatomical variables in the study of side veneer grafts of *Pinus engelmannii* Carr.

| Pairs of Variables Analyzed | Degrees of Freedom | Difference of Means | *t*-Value | Probability |
|-----------------------------|--------------------|---------------------|-----------|-------------|
| TAS vs TAR | 119 | −1.91 | −2.59 | 0.0107 |
| CAS vs CAR | 119 | 10.34 | 9.25 | <0.0001 |
| DRCS vs DRCR | 119 | 0.12 | 8.94 | <0.0001 |

TAS = total area of the scion; TAR = total area of the rootstock; CAS = area of the cambium in the scion; CAR = area of the cambium in the rootstock; DRCS = density of resin channels in the scion; DRCR = density of resiniferous channels in the rootstock.

The vascular cambium of the scion was irregular in shape, while the rootstock cambium was usually almost circular in shape (Figure 3).

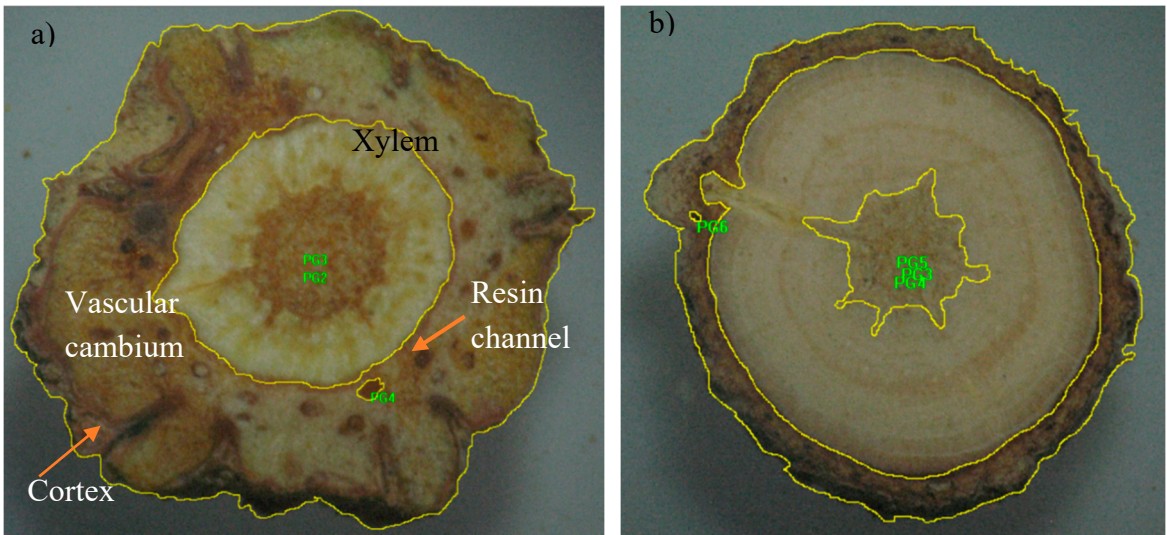

**Figure 3.** Anatomical sections of grafted organs of *Pinus engelmannii* Carr., observed at a total magnification of 40×: (**a**) scion, (**b**) rootstock.

### 3.2. Survival of Grafts

Graft survival rates six months after grafting were 12.5%, 32.5%, and 22.5% for the young, middle-aged, and old trees, respectively. The DTA was the only treatment that yielded significant differences ($p < 0.05$) (Table 6).

**Table 6.** Analysis of variance of survival of side veneer grafts of *Pinus engelmannii* Carr.

|  | Degrees of Freedom | Mean Square | F Value | *p* > F |
|---|---|---|---|---|
| Age class of the scion donor tree (DTA) | 2 | 0.133 | 8.11 | **0.015** |
| Scion provenance (SP) | 1 | 0.066 | 4.02 | 0.085 |
| Phenological stage of buds (PSB) | 1 | 0. 066 | 4.02 | 0.085 |

A substantial increase in mortality of grafts was observed through the six-month evaluation period, in the three DTA evaluated; however, the buds coming from the middle-aged trees yielded, finally, the highest survival of grafts (Figure 4).

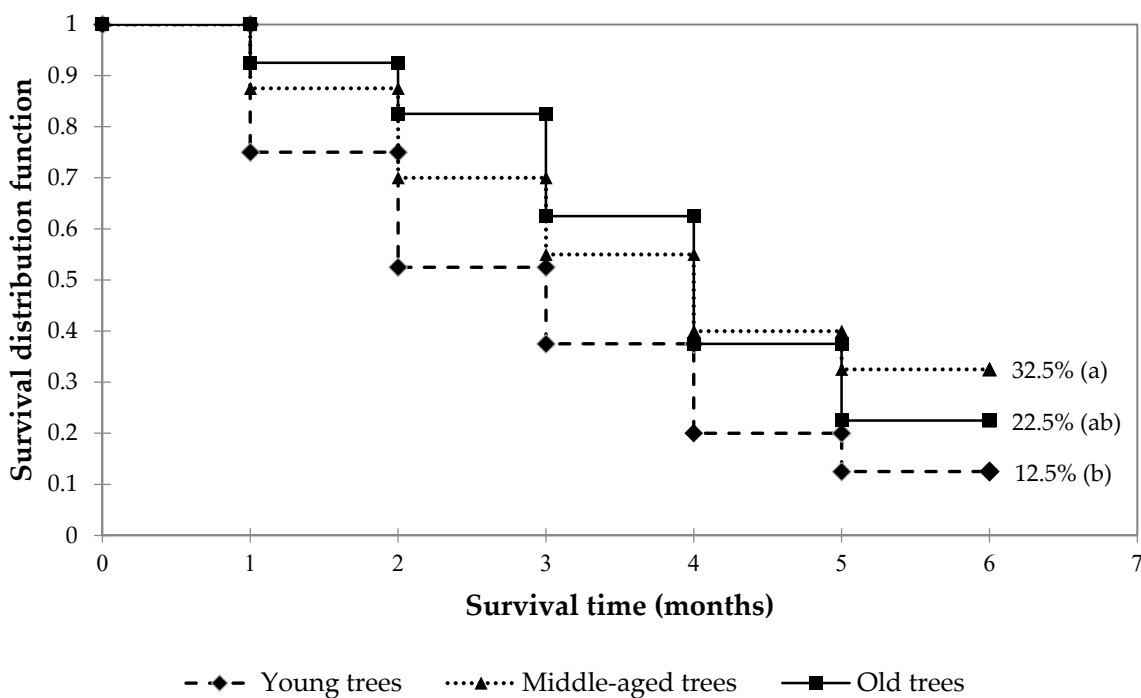

**Figure 4.** Survival function estimated with the Kaplan–Meier model for *Pinus engelmannii* Carr. grafts, made with buds coming from different age classes of the scion donor trees (DTA). Values followed by different letters indicate statistical differences according to the Tukey test ($p < 0.05$).

Throughout the evaluation period, besides registering the dynamics of grafts survival for the three age classes of the scion donor trees (DTA), the dynamics were also registered for the two phenological states of the buds (PSB) under study. The first three numerical columns in Table 7 show that the numbers of elapsed days before survival were below 75%, 50%, and 25% in each combination of DTA and PSB. The last column shows the final percentage of live grafts of each combination, at the end of the six-month evaluation period.

**Table 7.** *Pinus engelmannii* graft survival dynamics for each combination of age class of the scion donor tree (DTA) and phenological state of buds (PSB).

| Tree Age Class (Phenology) | Days Elapsed for Survival Less than the Indicated % | | | |
|---|---|---|---|---|
| | 75% | 50% | 25% | Final Survival (%) |
| Young (end of dormancy) | 30 | 90 | 150 | 20.0 |
| Young (beginning of sprouting) | 60 | 60 | 120 | 5.0 |
| Middle-aged (end of dormancy) | 90 | 120 | >150 | 45.0 |
| Middle-aged (beginning of sprouting) | 60 | 90 | 120 | 20.0 |
| Old (end of dormancy) | 90 | 120 | 150 | 15.0 |
| Old (beginning of sprouting) | | | | 30.0 |
| Total | 60 | 90 | >150 | 22.5 |

The Pearson correlation analysis, applied to the whole number of performed grafts, detected high multicollinearity (greater than 0.75) between the TAS, TAR, CAS, and CAR variables (Table 8), indicating that some of the variables should probably be considered as redundant when fitting the survival and growth models.

**Table 8.** Correlation matrix of anatomical variables of scions and rootstocks, considering all of the performed grafts.

| Variable | TAS | TAR | CAS | CAR | DRCS | DRCR | TAS/TAR | CAS/CAR | DRCS/DRCR |
|---|---|---|---|---|---|---|---|---|---|
| TAS | 1.00 | 0.95 | 0.92 | 0.76 | −0.59 | −0.55 | 0.24 | −0.17 | 0.10 |
| | | <0.0001 | <0.0001 | <0.0001 | <0.0001 | <0.0001 | 0.008 | 0.04 | 0.23 |
| TAR | **0.95** | 1.00 | 0.86 | 0.81 | −0.67 | −0.56 | −0.05 | −0.26 | 0.13 |
| | **<0.0001** | | <0.0001 | <0.0001 | <0.0001 | <0.0001 | 0.55 | 0.003 | 0.13 |
| CAS | **0.92** | **0.86** | 1.00 | 0.70 | −0.65 | −0.50 | 0.26 | −0.04 | 0.07 |
| | **<0.0001** | **<0.0001** | | <0.0001 | <0.0001 | <0.0001 | 0.003 | 0.65 | 0.39 |
| CAR | **0.76** | **0.81** | **0.70** | 1.00 | −0.52 | −0.36 | −0.05 | −0.66 | 0.01 |
| | **<0.0001** | **<0.0001** | **<0.0001** | | <0.0001 | <0.0001 | 0.51 | <0.0001 | 0.85 |
| DRCS | −0.59 | −0.67 | −0.65 | −0.52 | 1.00 | 0.49 | −0.20 | 0.14 | 0.22 |
| | <0.0001 | <0.0001 | <0.0001 | <0.0001 | | <0.0001 | 0.02 | 0.11 | 0.01 |
| DRCR | −0.55 | −0.56 | −0.50 | −0.36 | 0.49 | 1.00 | −0.01 | 0.003 | −0.64 |
| | <0.0001 | <0.0001 | <0.0001 | <0.0001 | <0.0001 | | 0.89 | 0.96 | <0.0001 |
| TAS/TAR | 0.24 | −0.05 | 0.26 | −0.05 | −0.20 | −0.01 | 1.00 | 0.27 | −0.11 |
| | 0.008 | 0.551 | 0.003 | 0.51 | 0.02 | 0.89 | | 0.002 | 0.20 |
| CAS/CAR | −0.17 | −0.26 | −0.04 | −0.66 | 0.14 | 0.003 | 0.27 | 1.00 | 0.06 |
| | 0.049 | 0.003 | 0.65 | <0.0001 | 0.11 | 0.96 | 0.002 | | 0.46 |
| DRCS/DRCR | 0.10 | 0.13 | 0.07 | 0.01 | 0.22 | −0.64 | −0.11 | 0.06 | 1.00 |
| | 0.23 | 0.137 | 0.39 | 0.85 | 0.01 | <0.0001 | 0.20 | 0.46 | |

TAS = total scion area; TAR = total rootstock area; CAS = area of the cambium in the scion; CAR = area of the cambium in the rootstock; DRCS = density of resin channels in the scion; DRCR = density of resiniferous channels in the rootstock.

Also, a stepwise regression was run in order to select the most significant independent variables, as well as the ratios between the anatomical variables of the grafted organs (TAS/TAR, CAS/CAR, DRCS/DRCR), to be included in the Cox proportional hazards model, in relation to the risk of death (to evaluate survival) of the grafts. This procedure detected that the better Cox survival prediction model is the model that includes only four variables, which yields the lowest AIC value (653.61) (Table 9).

**Table 9.** Results of the stepwise procedure for selecting the variables to be included in the Cox proportional hazards model, in relation to the risk of death of grafts.

| Variables Included | Number of Variables | AIC Value |
|---|---|---|
| DTA, SP, PSB, TAS, DRCS, DRCR, TAS/TAR, CAS/CAR, DRCS/DRCR | 9 | 660.6 |
| DTA, SP, PSB, TAS, DRCS, DRCR, CAS/CAR, DRCS/DRCR | 8 | 658.6 |
| DTA, SP, PSB, TAS, DRCS, DRCR, DRCS/DRCR | 7 | 656.9 |
| DTA, SP, TAS, DRCS, DRCR, DRCS/DRCR | 6 | 655.4 |
| DTA, SP, TAS, DRCS, DRCS/DRCR | 5 | 653.9 |
| DTA, SP, TAS, DRCS | 4 | 653.61 |

DTA = age category of the budding tree; SP = provenance of scions; PSB = phenological status of the buds; TAS = total scion area; DRCS = density of resin channels in the scion; DRCR = density of resiniferous channels in the rootstock.

Of the four variables that were chosen by using the stepwise procedure to adjust the proportional hazards model of Cox, only three (DTA, TAS, and DRCS) had a significant effect on graft survival ($p < 0.05$), therefore, the best model was considered to be that which was fit with only these three variables (Table 10).

**Table 10.** Cox proportional hazards model in terms of the risk of death of *Pinus engelmannii* Carr. grafts.

| Parameter | Estimator | |Z| | $p > |Z|$ | Hazard Ratio |
|---|---|---|---|---|
| Age class of the scion donor tree (DTA) | −0.2963 | −2.357 | 0.0184 | 0.1678 |
| Total area of the scion (TAS) | 0.0122 | 2.089 | 0.0367 | 0.1229 |
| Density of resiniferous channels in the scion (DRCS) | 3.1844 | 3.011 | 0.0026 | 0.0433 |

Given the negative value of the estimator of the DTA variable, grafts with scions from young trees are less likely to survive than grafts with scions from middle-aged or old trees (Table 10). The other two variables have positive value estimators, which indicate that the higher the TAS and/or DRCS, the lower the probability of graft survival (Table 10).

### 3.3. Effects on the Growth of the Surviving Grafts

Although the average growth at the end of the evaluation period varied between 43.4 and 38.6 mm, depending on the ages of the donor trees, no statistically significant differences were found between the three DTA levels, in relation to the growth of the grafts (Table 11).

**Table 11.** Mean value and standard error of sprouted graft growth, six months after grafting, by age group of scion donor trees (DTA).

| Age Class of the Scion Donor Tree (DTA) | Mean Growth (mm) | Number of Sprouted Grafts | Tukey Group * |
|---|---|---|---|
| Young | 43.4 ± 4.0 | 5 (12.5%) | a |
| Middle-aged | 38.6 ± 6.7 | 13 (32.5%) | a |
| Old | 42.9 ± 4.6 | 9 (22.5%) | a |

\* Same letters indicate no significant differences between treatments ($p < 0.05$).

Only one anatomical variable of the rootstocks (TAR) had a highly statistically significant correlation with the growth of the grafts (Table 12). With respect to the scion, the variables CAS and TAS showed a highly statistically significant correlation (more than 0.75) and with high significance ($p < 0.0001$) with respect to the longitudinal growth of the sprouted grafts, which were alive after six months (Table 12). However, those three variables show a high degree of multicollinearity (Tables 8 and 12); therefore, only the variable that showed the highest significant correlation (CAS) was included in the tested growth models (Table 12), since TAS became a redundant variable.

**Table 12.** Correlation matrix of the variables that could be used for growth prediction, taking into account only the grafts that were alive at the end of the six-month evaluation period.

| Variable | TAS | TAR | CAS | CAR | DRCS | DRCR | Graft Growth |
|---|---|---|---|---|---|---|---|
| TAS | 1.00 | **0.91** <0.0001 | 0.79 <0.0001 | 0.63 0.0004 | −0.74 <0.0001 | −0.49 0.008 | **0.77** <0.0001 |
| TAR | **0.91** <0.0001 | 1.00 | 0.73 <0.0001 | 0.66 0.0001 | −0.67 <0.0001 | −0.55 0.002 | **0.70** <0.0001 |
| CAS | **0.79** <0.0001 | **0.73** <0.0001 | 1.00 | 0.31 0.10 | −0.18 0.36 | −0.33 0.09 | **0.89** <0.0001 |
| CAR | 0.63 0.0004 | 0.66 0.0001 | 0.31 0.10 | 1.00 | −0.66 0.0002 | −0.16 0.39 | 0.29 0.12 |
| DRCS | −0.74 <0.0001 | **−0.67** <0.0001 | −0.18 0.36 | −0.66 0.0002 | 1.00 | 0.45 0.01 | −0.26 0.17 |

**Table 12.** *Cont.*

| Variable | TAS | TAR | CAS | CAR | DRCS | DRCR | Graft Growth |
|---|---|---|---|---|---|---|---|
| DRCR | −0.49 | −0.55 | −0.33 | −0.16 | 0.45 | 1−00 | −0.28 |
| | 0.008 | 0.002 | 0.09 | 0.39 | 0.01 | | 0.14 |
| Graft growth | 0.77 | 0.70 | 0.89 | 0.29 | −0.26 | −0.28 | 1.00 |
| | <0.0001 | <0.0001 | <0.0001 | 0.12 | 0.17 | 0.14 | |

### 3.4. Evaluation of Models Predicting Growth of Grafts

The four models evaluated for predicting the growth of the grafts were found to be highly significant ($p < 0.0001$). However, not all parameters were significant ($p < 0.05$) when estimating these models (Table 13).

**Table 13.** Parameters of the regression models that were fitted for predicting the longitudinal growth of grafts as a function of the scion cambium area (CAS).

| Model | $p < F$ (Model) | Parameter | Value | $p > \|t\|$ | Standard Error | AIC |
|---|---|---|---|---|---|---|
| Quadratic model $y_i = \beta_0 + \beta_1 x_i + \beta_2 x_i^2$ | <0.0001 | $\beta_0$ $\beta_1$ $\beta_2$ $R^2$ | −33.665 2.508 −0.017 0.814 | 0.223 0.050 0.200 | 26.954 1.216 0.013 | −71.777 |
| Logarithmic model $y_i = \beta_0 + \beta_1 ln x_i$ | <0.0001 | $\beta_0$ $\beta_1$ $R^2$ | −111.281 40.481 0.812 | <0.0001 <0.0001 | 14.622 3.887 | −147.398 |
| Linear model $y_i = \beta_0 + \beta_1 x_i$ | <0.0001 | $\beta_0$ $\beta_1$ $R^2$ | 1.408 0.913 0.801 | 0.725 <0.0001 | 3.970 0.091 | −71.898 |
| Exponential model $y_i = \beta_0 e^{\beta_1 x_i}$ | <0.0001 | $\beta_0$ $\beta_1$ $R^2$ | 16.320 0.021 0.782 | | 1.606 0.002 | −144.738 |

According to the fits obtained (Table 13), the CAS variable is directly proportional to growth of *Pinus engelmannii* grafts, as represented in Figure 5, which corresponds to the best-fit model.

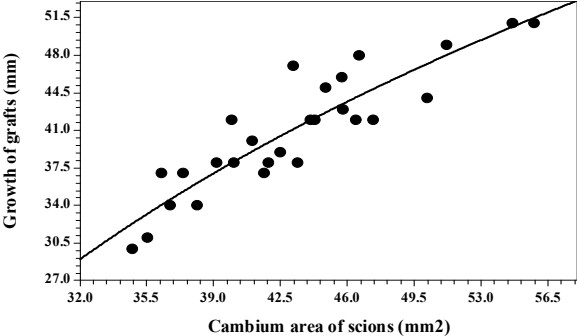

**Figure 5.** Scatter plot of the fit of the logarithmic model that was used to predict the longitudinal growth of grafts as a function of the cambium area on the scion (CAS).

In all cases, the coefficient $\beta_1$ was positive so that as CAS increased, the growth of the grafts also increased. The logarithmic model most accurately predicted the growth of the grafts (Table 13), since it has the smallest AIC value and both parameters ($\beta_0$ and $\beta_1$) were highly significant ($p < 0.0001$).

## 4. Discussion

In this study, the grafting was carried out at the end of the bud dormancy period, and at the beginning of bud sprouting. Relatively low survival results, after six months of grafting, were obtained for both phenological stages (Table 7). However, some authors have indicated that the season influences the success of grafting [7–9]. In a study of grafts of *Pinus patula* Schiede ex Schltdl. *et* Cham. in the state of Mexico, in South-Central Mexico, a survival rate of 34% was reported for seven-month-old grafts, for grafting performed at the end of winter (February), corresponding to the end of the dormant bud stage [12]. However, the findings of another study indicate that it is possible to successfully graft woody species from any part of the world at any time of the year under controlled climate conditions, as long as the maximum temperature is not higher than 24 °C and the minimum temperature is not lower than 3 °C [9]. For the grafting of *Araucaria angustifolia* (Bertol.) Kuntze in Parana, Brazil, in April (i.e., autumn, with scions in the dormant stage) with average temperatures that fluctuated between −3 °C and 18 °C and with a high relative humidity, a survival of 60% was reported at 12 months after grafting [7]. In our study, the survival decreased significantly after the second month of evaluation (March) onwards (Figure 4), which is attributed to the continual fluctuations in temperature (up to 42.59 °C) that were detected, and the low relative humidity that was registered within the greenhouse (38% on average). These results appear to be consistent with a different study [8], which recommends grafting during the colder months to avoid high temperatures, and possible dehydration of the scion–rootstock junction. However, in a former *P. engelmannii* grafting experiment in Durango, Mexico, 38.6% survival was obtained when grafting in February [44]. In the same study, in *P. cooperi* C.E. Blanco grafts, 57% survival was reported when grafts were made in February, and 65% survival was reported when grafting in March [44].

The highest survival of grafts of the species in the present study, was obtained with the scions that were collected from 20–60-year-old trees (Figure 4). A similar situation was reported in an experiment with *Araucaria angustifolia*, in which it was possible to successfully graft 33-year-old tree scions [7].

In the present research, the age class of the scion donor tree (DTA) had a significant effect on survival (Figure 4 and Table 10), but not on graft growth (Table 11). In a different study, it was found that *Abies fraseri* (Pursh) Poir. grafts, that were established with young tree scions, grew better than grafts that were established with scions from old trees [45].

In the present study, no significant differences were found in the provenance of the scions in relation to the survival of the grafts, since they resulted in being statistically similar (Table 6). In another study using scions from four different provenances in the Argentine Patagonia, the provenance did not affect the survival of grafts of *Austrocedrus chilensis* (D. Don) [46]. However, the provenance of the scions was found to influence the survival of *Pinus patula* grafts, in a study within which scions were collected from two different provenances in the state of Mexico [12]. However, due to the genetic variability among populations of *Pinus* species, it is advisable to produce rootstocks with seed from the same areas where the scions will be collected [47]. In addition, the age of the rootstocks can be a determinant in the success of the graft. Therefore, it is advisable to use rootstocks of two to three years of age, because the older the organs to be grafted are, the higher the biological and genetic specialization of cell tissues, which increases the difficulty of vegetative propagating of *Pinus* species [48].

According to the results of stepwise regression, the shape and proportion of the areas of the cuts in the scions and the rootstocks (TAS/TAR and CAS/CAR) did not significantly affect the graft death or survival rates (Table 9). In a similar manner, the total area of the rootstock (TAR) was higher than TAS, and CAS was higher than CAR (Table 5), but these relationships among anatomical variables were not found to have significant effects in the studied response variables (Table 9). These findings are consistent with those reported for grafts of the *Prunus* genus, in which the area of the scions and rootstocks did not affect graft survival [49]. Also, when *Pinus patula* barbs were grafted into rootstocks of *P. douglasiana* Martínez and *P. pseudostrobus* Lindl., the area of the scions being smaller than the area of the rootstocks did not affect the success of top cleft grafts [12]. However, when grafting *Cynara*

*cardunculus* var. *scolymus* L., it was found that the survival was significantly enhanced when the dimensions of the cut area of the scions and the rootstock plants were of similar size [50]. Similarly, the success of grafts of various *Litchi* species increased significantly when the sizes of the anatomical areas of both components of the graft were similar [51]. The results above suggest that the similarity in shape and size of the cuts of the scion and the rootstock improves the possibilities for better matching between cells of the same type in the grafted organs, as found in a study of *Pinus patula*, *P. greggii* Engelm. ex Parl. var. *australis* Donahue *et* Lopez, *P. leiophylla* Schiede ex Schltdl. *et* Cham. and *P. teocote* Schiede ex Schltdl., in which it was reported that the irregular shape of the anatomical structures generates incompatibility during grafting [36].

The effect of cambium has been reported in different graft survival studies [26,36,52]. It has been reported that the meristematic activity of the cambium between the scion and the rootstock is an important factor in the successful establishment of the graft [16,17]. In the present study, the size of the surface area of the scions and the rootstock plants showed a similarity of 97%; however, the cambium of the scions is irregularly shaped, in contrast to the almost circular shape of the cambium of the rootstock plants (Figure 3). Such a mismatch between the shapes of the exposed cambium of the scions and rootstocks may have made the contact of the ducts difficult, thus hindering callus formation [3]. This may have contributed to the low rate of graft survival that was observed in our study.

In previous research, it was found that the resin did not have a negative effect on the grafting of *Anacardium occidentale* L. Further, in the same study, it was reported that the presence of resin enables a temporary union between the scion and the rootstock [24], which would be favored by abundant secretion channels. However, in the present study, we found that when there was a higher density of resiniferous channels in the scion, there was also a higher mortality of *Pinus engelmannii* grafts (Table 10), although this might be related to the significant difference in the number of channels between the scion and the rootstock (Table 5), which could make adequate connection between the channels of the two grafted organs difficult.

In general, it has previously been noted that graft survival mainly depends on the correct execution of the grafting technique, in addition to the anatomical characteristics of the scions and rootstocks [1,17]. In that sense, our study findings represent an advance in the knowledge of the topic. However, this fact should be complemented by further studies of compatibility and incompatibility of grafts in *P. engelmannii* and other species of the genus *Pinus* and conifers in general, depending on their genetic, phenological, anatomical, and histological characteristics.

## 5. Conclusions

Although the survival rate of side veneer grafts of *Pinus engelmannii* was relatively low compared to those reported in other studies for other species, the grafting was considered to be successful, as vegetative propagation of this species is difficult when utilizing this technique. Grafting with buds from *P. engelmannii* in two different phenological stages made a statistically non-significant difference, as the survival rates were similar when grafting, with buds at the end of the dormant stage and at the beginning of budding activity. Graft survival did not depend on either of the two geographical provenances of the scions. On the other hand, the age class of the budding tree significantly affected graft survival, and the best scions for grafting were those from donor trees aged between 20 and 60 years (middle-aged trees).

It was found that some anatomical variables can be used to predict the survival rates, since the graft survival rate tended to decrease as the total area and the density of resin channels of the scion increased. However, the same anatomical variables in the rootstocks did not seem to influence the survival of *P. engelmannii* grafts. In a similar manner, the effect of the cambium on the scion and the effect of the rootstock on the survival and growth of the grafts were evident.

**Author Contributions:** A.P.-L. designed the experiment, collected and analyzed the data and wrote the manuscript; J.C.H.-D. participated in designing and establishing the experiment, analyzing the data and revising the manuscript; J.A.P.-R. contributed to designing and establishing the experiment and revising the manuscript;

J.L.-U. provided advice on the treatments and also revised the manuscript; A.C.-P. participated in analyzing the anatomical structures and revising the manuscript; C.W. participated in analyzing the data and in revising the manuscript; J.A.C.-S. contributed to analyzing the anatomical structures and revising the manuscript.

**Funding:** Consejo Nacional de Ciencia y Tecnología: 441054Consejo de Ciencia y Tecnología del Estado de Durango: COCYTED-12/02/18/265.

**Acknowledgments:** The authors are grateful to M.C. Santiago Solís González for the support provided harvesting the vegetative material used in the present study. The authors are also grateful to the National Council of Science and Technology (CONACYT) for financial support (via a fellowship) for the first author of this article, as well as to the Council of Science and Technology of the State of Durango (COCYTED), for the financial support via research project on coniferous grafts.

**Conflicts of Interest:** The authors declare no conflict of interest.

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
