# Peer review of "Some Factors Involved in the Success of Side Veneer Grafting of Pinus engelmannii Carr."

_forests, doi:10.3390/f10020112_

Round 1

Reviewer 1 Report

The paper needs a major general revision to be accepted.

The title should describe the trial with more accuracy. Not only anatomic and phenological traits were studied/considered, an important factor was also the age of donor tree, for example. The title must describe the work made. For example: 'Factors involving the graft sucess of  ....'

Abstract oK

The introduction does not frame the interest of your work. There is no justification of the objective to graft this species; Why is this Pinus important in Mexico, for example? Genetic preservation? Production?  The information provided is more suitable for discussion rather than for the introduction.

M&M: Why do you chose these two origins? Which are the differences between sites? How do you select the trees?

In line 99 'selected' must be changed by 'collected'

To get conclusions of grafting success to take only 10 individuals per treatments is a very low number.

Table 2, data of climatic conditions in the greenhouse it is not necessary, saying that you controlled temperature and RH it is enough.

Anatomical data studied are the most suitable ones and the statistical analysis used too. The problem is the low number of data you are analyzing.

Results: In Figure 3, the magnification must be indicated.

Table 7 is not necessary if you include Figure 4. The signification level could be showed in the Figure. Perhaps it would be clearer to use percentage of survival also in the Figure.

To modelise the answer to the graft much more data is necessary.

Author Response

Answers letter to Reviewer 1.

We greatly appreciate your comments and suggestions made in your review of our manuscript Anatomy and Phenology of Scions and Rootstocks in Relation to the Survival and Growth of Pinus engelmannii Carr. Grafts”. Below, we describe our answers and the changes made to our MS, trying to attend your observations.

1.       Title of the paper

Answers:

Following your suggestion, the title was restructured to give greater support to the presentation of the article. Now is read asSome factors involved in the success of side veneer grafting of Pinus engelmannii Carr.

2.       Introduction

Answers:

More literature was reviewed in order to improve the introduction, trying to give more support and justification to the objectives of our study. First, the importance of grafting is stated, pointing out its objectives. Then, the importance of the studied species was described. Likewise, we reordered some texts and improved redaction, to give a better sequence to the concepts included in this section. The changes are marked with color in the MS.

3.       Materials and methods

Answers:

The specified better why the two scion collect provenances were selected and gave additional information about each provenance. The donor trees where selected by age; visually we picked the young, middle aged and old trees in each location and then counted the growth cores to make sure the corresponded to the age ranges to be compared.

In line 99 we corrected the word “selected” with “collected”.

In regard to your observation on the number of individuals per treatment, we are not able to make any changes at this stage. However, we think is important to mention that, at the time when we developed this first experiment we were unable to get enough adequate grafting material (rootstocks) to perform a larger number of grafts in each treatment. We analysed such situation and decided that 10 repetitions could be a minimum, to be able to perform valid statistical analyses, specially using not parametric techniques, which we used.

Table 2 was deleted, since it was not necessary (as you suggested) and following also one suggestion of Referee 2. We only stated that we recorded temperature and relative humidity data along the duration of the experiment.

Thanks for your support on our selection of anatomical data to be collected and the statistical analyses used.

4.       Results

Answers:

We suppressed Table 7, as you suggested and instead improved figure 4 adding the percentage of survival in each age category.

Best regards and many thanks for your time and attention.

PhD. José Ciro Hernández Díaz

Reviewer 2 Report

The present paper is devoted to an extremely interesting and not so often studied problem, namely, what factors affect vegetative propagation of conifers through grafting and how this is based on anatomical (and physiological) characteristics. However, the experiments seems to be not properly planned and performed, and the paper itself is not well written. The authors heavily rely on mathematical analysis forgetting to show and describe biological effects. As a result, the paper loses its scientific significance and degree of generalization of the obtained results.

Introduction

Not straightforward, lacking logical sequence. Need to start with explanation why vegetative propagation of Pinus is necessary. The first two sentences from the Abstract can be used as an initial thought. More specific particular problems need to be analyzed further, leading to the aim of the study.

Uncorrect use of many references is evident in many places. For example, [4,5] (line 39) are not correct, as study [4] analyzes western larch instead of comparing Pinus species, but study [5] studies only one Pinus species. References [6–10] seems to be not the best sources for describing factors affecting grafting success, some good reviews need to be mentioned here instead. [29] and [30] refers to a signle source. It seems that the authors artificially accomodate references to their ideas.

There is an incorrect description of grafting, pointing to aerial and underground parts, but both parts are usually connected above ground, and different types of grafting can involve only small branches, buds on small twigs etc.

It is not becoming clear what is meant by "catalytic complex" (line 68). Is it group of cells acting as initials of callus?

Materials and methods

The authors seems to be not sure about the exact physiological phase of their grafting material. It is mentioned in the M&M part that "in February (13 and 14) 2017 (buds in the dormant state) and in March (20 and 21) 2017 (sprouting buds)", but further in the Discussion it is said that "grafting with scions at the end of the dormant stage and at the beginning of budding activity". This is clearly contradictory. Both sampling dates seem to be too close to make exact destinction between the states.

Another methodological problem is related to the number of separate experiments. As there were two scion collection dates, and grafts were performed on the same day, it seems that there were two separate experiments with six treatments each, not a single one with 12 treatments as it is mentioned. As rootstock plants also underwent development within more than a month period, it is incorrect to directly compare these data. Also, why 5 to 7 year-old plants were used as rootstocks, not with the same age? It also could grately affect the results.

In Table 2, column indicated as "Date" shows some kind of numbers from 7 to 35 with month of brackets. Seem that these numbers represent weeks? In fact, information from Table 2 has not been used further and can be avoided.

Results

Most unfortunately, the results itself are not presented, described and analyzed in full. Therefore, validity of conclusions are based only on some type of mathematical analysis, but one cannot judge on correctedness of the procedure without particular data. It is not clear why the authors concentrate on "death" (as qualitative measurement) instead of success of grafting in quantitative sense? The authors already mention that "The variables CAS and TAS of the scions were highly significantly correlated with the longitudinal growth of the sprouted grafts. However, only the variable that showed the highest highly significant correlation (CAS) was included in the growth models, because both variables show a high degree of autocorrelation (p < 0.0001).", which points to serious problems with their mathematical models on the basis of complete lack of functional analysis.

Discussion

Mostly useless without actual data.

Author Response

Answers letter to Reviewer 2.

Thanks a lot for taking part of your time to make observations to our manuscript Anatomy and Phenology of Scions and Rootstocks in Relation to the Survival and Growth of Pinus engelmannii Carr. Grafts”. Below, we describe our answer and the changes made to our MS.

Title of the MS. Following a suggestion of Referee 1, the title was modified to better support the presentation of the article. Now is read as:Some factors involved in the success of side veneer grafting of Pinus engelmannii Carr.”

We also made an effort to complement our mathematical-statistical analyses with biological and environmental considerations, trying to improve the interpretation of our results.

Answers to your observations

1.       Introduction

Answers:

Seeking to improve the introduction, we reviewed more literature and also, we checked and corrected the use of some references that were already had included in the whole document. Trying to attend your observation we reordered some texts and improved redaction of our work. First, we described the importance of grafting and vegetative propagation in general. Then, we described some antecedents from other studies on aspects relevant to our study. Later, we talk about the important use of grafting in the establishment of asexual seed orchards of conifers and also describe the importance of the studied species, seeking to more directly lead to the aim of our study.

All changes are marked with yellow color in the MS.

We cited [29] and [30], (now [22] and [23]), as separate sources because they are works of different authors, although both studies are included in de same book, along with other studies.

You are right, grafting may involve just aerial portions of the plants. Therefore, we corrected our description of grafting. At the beginning of the introduction (lines 36-37), now is read as: “Grafting is a vegetative propagation technique that consists of joining parts of two different plants (a scion with a rootstock), which fuse together forming a callus and then, grow as a single plant [1–3]”.

Lines 63-64. We clarified the definition of “catalytic complex” as an agglomeration of meristematic cells [20,21].

2.       Materials and methods

Answers

Lines 108-111: Thanks for the observation. The two phenological stages of the buds are now better defined. In these lines we clarified by saying: “…two collects were made to obtain scions with buds at two phenological stages, in February (13 and 14) 2017 (buds considered to be at the end of dormant state, since the colder months are December and January) and in March (20 and 21) 2017 (buds visually at the beginning of sprouting)”. Also, we took care of being consistent in the other section of our MS.

Lines 124-131. Your kind observation about the range of rootstock ages (5-7 years) and the two collect dates of the scions, as well as their possible effect on the results of our experiment, allowed us to better explain our reasoning in that respect, and also to explain why we considered this work as a single experiment. We added lines 124-131 to explain such reasoning by saying: “We used rootstocks of five to seven years instead of just one age, because their variability in diameter allowed a better matching with the dimensions of the scions, which were also variable despite having taken, all of them, from one-year branchlets. Besides, since the grafting was done at the end of the winter season and the age of the rootstock was not a variable to evaluate, we considered negligible the possible growth undergone by the rootstock between the two grafting dates, which certainly were considered an independent variable to evaluate the success of grafting, because the phenological stage of the buds obviously changed”.

Following your suggestion, we deleted Table 2 from our document, because it was not necessary.

3.       Results

Answers:

In attention to your very appropriate observations, on giving more elements to judge the validity of our discussion and conclusions, we made several improvements, as follows:

Lines 270-271: We added “Table 7. Pinus engelmannii graft survival dynamics by age class of the scion donor tree and phenological state of buds.”, as well as the explanation of its content.

Lines 276-280: We added “Table 8. Correlation matrix of anatomical variables of scions and rootstocks, considering all the performed grafts.”, and its explanation.

Lines 282-292: We improved the explanation of Table 9, which shows the “Results of the stepwise procedure for selecting the variables to be included in the Cox proportional hazards model, in relation to the death risk of grafts”.

Lines 293-297. We improved the explanation of “Table 10. Cox proportional hazards model in terms of survival of Pinus engelmannii grafts”.

Lines 302-305. Improved description of the results shown in Table 11

Lines 317-319. We included “Table 12. Correlation matrix of the variables that could be used for growth prediction, taking into account only the grafts alive at the end of the six-month evaluation period”, and its description.

Lines 321-326. Improved explanation of Table 13 (which was before Table 11), containing Parameters of the regression models fitted for predicting the longitudinal growth of grafts as a function of the scion cambium area (CAS).

In Figure 5, we left just one diagram (instead of four that we had before), showing the positive correlation to predict the longitudinal growth of grafts as a function of the area of the cambium on the scion (CAS).

In regard to your question: It is not clear why the authors concentrate on "death" (as qualitative measurement) instead of success of grafting in quantitative sense? Let me say that, we start talking about “death of the grafts” because this term is more appropriate to describe “risk of something” (in this case risk of death), which is consistent with the type of phenomena that usually are analysed with the “Cox proportional hazards model”. However, you are right, because our main interest should be the description of the possibilities of success in the survival and growth of the grafts, and this is why we present and discuss Table 10, which shows the Cox proportional hazards model in terms of the survival of Pinus engelmannii grafts.

Regarding your kind observation about the high significance of CAS and TAS in relation to the longitudinal growth of the sprouted grafts, and why we latter base our discussion just considering the most highly correlated variable (CAS). In this respect now we tried to explain in our document that some independent variables showing high significant correlation between them (multicolinearity) probably should be excluded from the respective models, if just one of such highly correlated variables can represent the effect of the other in the model (Lines 309-315). In that sense we left in the growth model, just the variable most correlated with growth (CAS), since TAS in this case was considered to be a redundant variable.

4.       Discussion and conclusions

In these two sections we made several changes, trying to improve them based on the improvements made in our Results section (already described above). It would be impractical to include in this letter a detailed description of such changes, however they are highlighted in colour in the text of our MS (Lines 325 to the end of the document)

Dear Referees, thanks again for your observations and suggestions which were very valuable for improving our manuscript, which we intended to do to the best of our knowledge and given the short period of time allowed to make the improvements.

Best regards.

PhD. José Ciro Hernández Díaz

Round 2

Reviewer 1 Report

The article has been clearly improved.

There is a mistake from the line 87 to 90, the text is writing in Spanish!!!

The reference 34 does not exist in the text.

Author Response

Reviewer 1

Response letter

Thanks a lot for your new observations:

You can see that the new main changes to our document are now highlighted in blue.

In regard to the research design, at this stage we cannot make changes. However, we tried to improve our explanation of the way how we analysed our data (line 128).

Lines 88-91. Thanks for your observation, we translated those lines to English.

Line 84. The reference [34] is included in our citation… [19,33-37].

Thanks again for your attention.

Reviewer 2 Report

The manuscript has been greatly improved. Especially, methodological problems were discussed in detail, allowing for better understanding of possible generalization of the results.

Author Response

Reviewer 2

Response letter

In regard to your recommendation of checking the correct use of English, we are glad to inform that we sent our manuscript to a professional reviewer of papers, who is a native English speaker. He has made this service for us also in other occasions.

Let me also inform you that the main changes and improvements that we made to our document in this second round, are highlighted in blue, except the changes referred to the improvement of the use of English

Thanks a lot for your observation in the sense that our manuscript was improved. This was much possible due to your deep and wise observations in your first reviewing of our MS.
